# Structure-informed Risk Minimization for Robust Ensemble Learning

**Fengchun Qiao** [1]  **Yanlin Chen** [1]  **Xi Peng** [1]

## Abstract

Ensemble learning is a powerful approach for improving generalization under distribution shifts, but its effectiveness heavily depends on how individual models are combined. Existing methods often optimize ensemble weights based on validation data, which may not represent unseen test distributions, leading to suboptimal performance in out-of-distribution (OoD) settings. Inspired by Distributionally Robust Optimization (DRO), we propose Structure-informed Risk Minimization (SRM), a principled framework that learns robust ensemble weights without access to test data. Unlike standard DRO, which defines uncertainty sets based on divergence metrics alone, SRM incorporates structural information of training distributions, ensuring that the uncertainty set aligns with plausible real-world shifts. This approach mitigates the over-pessimism of traditional worst-case optimization while maintaining robustness. We introduce a computationally efficient optimization algorithm with theoretical guarantees and demonstrate that SRM achieves superior OoD generalization compared to existing ensemble combination strategies across diverse benchmarks. Code is available at: `https://github.com/deep-real/SRM`.

## 1. Introduction

Ensemble learning has emerged as a promising approach for out-of-distribution (OoD) generalization (Lee et al., 2023; Pagliardini et al., 2023), demonstrating superior robustness and adaptability compared to individual models when deployed in environments that differ from their training conditions. By combining multiple diverse models, ensemble methods can capture different aspects of the data distribution and reduce the impact of individual model biases (Wortsman et al., 2022; Ramé et al., 2023), leading to more reliable predictions under distribution shifts.

The success of ensemble learning hinges on two key steps: (1) training diverse individual models and (2) optimally combining them into a final predictor. While significant advances have been made in ensemble training to produce diverse and complementary models (Benoit et al., 2024; Pagliardini et al., 2023; Rubinstein et al., 2024), poor combination strategies can severely degrade performance (Rame et al., 2022). This underscores the importance of determining optimal model combinations. Existing methods fall into four main categories: selection/pruning (He et al., 2024), voting (Tsoumakas et al., 2008), weighted voting (Wu et al., 2023), and stacking (Chatzimparmpas et al., 2020). These approaches typically optimize ensemble weights based on validation data, either from a training split or a holdout set. However, due to distribution shifts, such learned combinations may underperform on unseen test distributions. A recent work (Qiao & Peng, 2024) attempted to mitigate this issue by explicitly promoting prediction diversity on test data. However, test data are typically unavailable in OoD generalization. This raises a fundamental question: *Can we learn ensemble weights that effectively generalize to unseen test distributions?*

Distributionally Robust Optimization (DRO) (Shalev-Shwartz & Wexler, 2016), originally designed for training individual models under distribution shifts, offers a promising approach to learning robust ensemble weights. DRO optimizes the worst-case risk over an uncertainty set of potential test distributions, typically defined as a divergence ball around the training distribution using distance metrics like $f$-divergence (Namkoong & Duchi, 2016), Wasserstein distance (Shafieezadeh Abadeh et al., 2018) or Maximum Mean Discrepancy (Staib & Jegelka, 2019). While DRO enhances robustness, it often leads to overly pessimistic solutions (Hu et al., 2018; Frogner et al., 2021; Dai et al., 2023), degrading average performance by considering implausible distributions that rarely occur in practice. This issue arises because DRO treats distributions in the uncertainty set independently, ignoring the intrinsic structure governing their real-world relationships.

To overcome this limitation, we propose Structure-informed Risk Minimization (SRM), a principled framework that

[1]DeepREAL Lab, Department of Computer and Information Sciences, University of Delaware, DE, USA. Correspondence to: Xi Peng <xipeng@udel.edu>.

*Proceedings of the 42nd International Conference on Machine Learning*, Vancouver, Canada. PMLR 267, 2025. Copyright 2025 by the author(s).

leverages distributional graphs to optimize ensemble weights for OoD generalization. Our key insight is to incorporate structural relationships between distributions into the uncertainty set, capturing how distributions naturally vary and influence each other. By focusing on realistic distribution shifts rather than arbitrary worst-case scenarios, SRM retains the robustness benefits of DRO while mitigating its excessive conservatism.

Our main contributions are: (1) A principled framework for learning ensemble weights under distribution shifts, bridging a critical gap in ensemble learning for OoD generalization. (2) A novel method for constructing uncertainty sets that integrate distributional relationships, providing a more realistic approximation of potential test distributions. (3) A computationally efficient optimization algorithm that balances worst-case robustness with average performance, supported by theoretical guarantees.

## 2. Background

### 2.1. Problem Formulation

Let $(X, Y)$ be random variables where instances $x \in \mathcal{X} \subseteq \mathbb{R}^d$ and labels $y \in \mathcal{Y}$ follow an unknown joint distribution $P(X, Y)$. We observe data under different environmental conditions $e$, where samples are drawn from a set of distributions $\mathcal{E}_{\text{all}}$ such that $(x, y) \sim P_e(X, Y)$. Given $m$ pre-trained models $\mathcal{F} = \{f_1, ..., f_m\}$ trained on $\mathcal{E}_{\text{train}} \subsetneq \mathcal{E}_{\text{all}}$, where each model $f_i : \mathcal{X} \to \mathcal{Y}$ maps inputs to predictions, our objective is to construct an optimal ensemble that generalizes to unseen test distributions $\mathcal{E}_{\text{test}} = \mathcal{E}_{\text{all}} \backslash \mathcal{E}_{\text{train}}$.

We achieve this by learning ensemble weights $\mathbf{w} \in \Delta^m$ (the probability simplex) to form the predictor:

$$f_{\mathbf{w}}(x) = \sum_{i=1}^{m} w_i f_i(x). \tag{1}$$

For a distribution $P_e$, the expected risk is defined as: $\mathcal{R}_{P_e}(\mathbf{w}) = \mathbb{E}_{(x,y) \sim P_e}[\ell(f_{\mathbf{w}}(x), y)]$, where $\ell$ is the loss function. During deployment, the ensemble encounters test distributions from $\mathcal{E}_{\text{test}}$ that are distinct from $\mathcal{E}_{\text{train}}$. The goal is to minimize the worst-case risk across $\mathcal{E}_{\text{test}}$:

$$\min_{\mathbf{w} \in \Delta^m} \sup_{e \in \mathcal{E}_{\text{test}}} \mathcal{R}_{P_e}(\mathbf{w}). \tag{2}$$

The key challenge lies in determining the optimal ensemble weights $\mathbf{w}$ using only training distributions $\mathcal{E}_{\text{train}}$ while ensuring generalization to unseen test distributions $\mathcal{E}_{\text{test}}$.

### 2.2. Average Risk Minimization

A natural approach is Empirical Risk Minimization (ERM) (Vapnik & Vapnik, 1998), which assigns ensemble weights to minimize the *average* risk on $P_{\text{train}}$:

$$\min_{\mathbf{w}} \mathcal{R}_{P_{\text{train}}}(\mathbf{w}) = \min_{\mathbf{w}} \mathbb{E}_{(x,y) \sim P_{\text{train}}}[\ell(f_{\mathbf{w}}(x), y)]. \tag{3}$$

However, ERM assumes that the training and test distributions are identical, which rarely holds in practice. Consequently, ERM-weighted ensembles may reinforce spurious correlations that do not generalize (Arjovsky et al., 2019).

### 2.3. Worst-case Risk Minimization

Distributionally Robust Optimization (DRO) (Shalev-Shwartz & Wexler, 2016) offers an alternative by minimizing the *worst-case* risk over an uncertainty set $\mathcal{U}(P_{\text{train}})$:

$$\min_{\mathbf{w}} \max_{P \in \mathcal{U}(P_{\text{train}})} \mathcal{R}_P(\mathbf{w}). \tag{4}$$

The uncertainty set is often defined as a divergence ball centered at $P_{\text{train}}$:

$$\mathcal{U}(P_{\text{train}}) = \{P : D(P, P_{\text{train}}) \leq \rho\}, \tag{5}$$

where $\rho$ controls the set size, and $D(\cdot)$ denotes a discrepancy measure between probability distributions, such as $f$-divergence (Namkoong & Duchi, 2016), Wasserstein distance (Shafieezadeh Abadeh et al., 2018; Qiao & Peng, 2021), and other statistical distance measures.

While DRO enhances robustness, it often produces *overly conservative* solutions (Frogner et al., 2021; Sagawa et al., 2019) by treating all distributions within distance $\rho$ as equally likely, regardless of their practical relevance. This limitation arises because DRO considers each potential test distribution independently, ignoring the underlying structure that governs real-world distributional shifts.

## 3. Structure-informed Risk Minimization

We propose Structure-informed Risk Minimization (SRM), a principled framework that leverages distributional graphs to optimize ensemble weights for out-of-distribution generalization. The key innovation lies in incorporating the inherent relationships between distributions into uncertainty set construction, capturing how distributions naturally vary and influence each other. Using this structural information, SRM guides the uncertainty set to focus on distributions likely to occur under real-world conditions, rather than arbitrary shifts. This maintains the robustness benefits of DRO while mitigating its tendency toward excessive pessimism. SRM remains computationally tractable through a carefully formulated optimization problem that balances worst-case robustness with average performance.

### 3.1. Structure-informed Uncertainty Sets

Let $\mathcal{G} = (\mathcal{V}, \mathbf{A})$ be a weighted graph, where the set of vertices $\mathcal{V}$ corresponds to the $n$ training distributions in $\mathcal{E}_{\text{train}} = \{P_1, ..., P_n\}$. The edges between distributions are represented by the adjacency matrix $\mathbf{A} \in \mathbb{R}^{n \times n}$, where each entry quantifies the similarity between distributions.

Specifically, the edge weight between two distributions $P_i$ and $P_j$ is defined as: $A_{ij} = D(P_i, P_j)$, where $D(\cdot)$ is a distributional distance metric (*e.g.*, Wasserstein distance).

To incorporate this structure into ensemble learning, we use graph centrality measures (*e.g.*, closeness centrality) to refine the construction of the uncertainty set in robust optimization. Central distributions serve as "anchors" that better represent the structure of the training domain, guiding the ensemble weighting process. By considering centrality, we ensure that the uncertainty set focuses on plausible test distributions that are well-supported by training data, rather than arbitrarily distant shifts. This helps mitigate over-pessimism in worst-case optimization while improving generalization to unseen distributions.

**Definition 3.1** (Centrality Prior). Given a graph $\mathcal{G} = (\mathcal{V}, \mathbf{A})$, we define the prior $\mathbf{p} \in \Delta^n$ as:

$$p_e = \frac{c(P_e)}{\sum_{i=1}^n c(P_i)}, \quad e \in \{1, ..., n\}. \quad (6)$$

Here $c : P_e \to \mathbb{R}^+$ maps each distribution to a non-negative value measuring its centrality in $\mathcal{G}$.

The uncertainty set is then defined as a mixture of training distributions constrained by the prior $\mathbf{p}$:

$$Q = \left\{ \sum_{e=1}^n q_e P_e \mid \mathbf{q} \in \Delta^n, \mathcal{D}(\mathbf{q}\|\mathbf{p}) \leq \tau \right\}, \quad (7)$$

where $\mathcal{D}(\cdot)$ denotes the distance between two distributions (*e.g.*, $\ell_2$ distance or KL divergence), and $\tau$ controls the allowable deviation from the structural prior. This ensures the uncertainty set focuses on plausible mixtures aligned with the structural relationships among training distributions.

**Alternative Priors**. We consider two alternatives to our centrality-based prior: (1) *Uniform prior*, where $p_e = \frac{1}{n}$ for all distributions, ignoring the graph structure. (2) *Graph Laplacian-based prior*, which uses the constraint $\mathbf{q}^\top \mathbf{L} \mathbf{q} \leq \tau$, where $\mathbf{L} = \mathbf{D} - \mathbf{A}$ is the graph Laplacian, with $\mathbf{D} = \text{diag}(\mathbf{A1})$ being the degree matrix. Here, $A_{ij} = \exp(-D^2(P_i, P_j)/2)$ measures the similarity between distributions $P_i$ and $P_j$. Although the Laplacian constraint encourages smooth variations over the graph, it focuses on local smoothness rather than global influence. In contrast, our centrality-based prior directly measures global proximity, better aligning with our goal of identifying distributions informative about unseen test conditions. The empirical results in Sec. 4 validate this choice.

## 3.2. Optimization Framework

Given the uncertainty set $Q$, we formulate the ensemble weight optimization as a constrained minimax problem:

$$\min_{\mathbf{w} \in \Delta^m} \max_{\mathbf{q} \in \Delta^n} \sum_{e=1}^n q_e \mathcal{R}_{P_e}(\mathbf{w}) \quad \text{s.t.} \quad \mathcal{D}(\mathbf{q}\|\mathbf{p}) \leq \tau, \quad (8)$$

where $\mathcal{R}_{P_e}(\mathbf{w}) = \mathbb{E}_{(x,y) \sim P_e}[\ell(f_{\mathbf{w}}(x), y)]$ denotes the expected risk on distribution $P_e$. To address computational challenges in Eq. 8, we use the Karush-Kuhn-Tucker (KKT) conditions (Boyd, 2004) to derive a Lagrangian reformulation, where the constraint is replaced with a penalty term:

$$\min_{\mathbf{w} \in \Delta^m} \max_{\mathbf{q} \in \Delta^n} \sum_{e=1}^n q_e \mathcal{R}_{P_e}(\mathbf{w}) - \lambda \mathcal{D}(\mathbf{q}\|\mathbf{p}), \quad (9)$$

where $\lambda > 0$ controls the trade-off between risk minimization and structural consistency. We solve this optimization problem using an alternating gradient algorithm: (1) Updating ensemble weights $\mathbf{w}$ via:

$$\mathbf{w}^{t+1} = \mathcal{P}_{\Delta^m}(\mathbf{w}^t - \eta_{\mathbf{w}}^t \nabla_{\mathbf{w}} \mathcal{L}(\mathbf{w}^t, \mathbf{q}^t)). \quad (10)$$

(2) Updating mixture weights $\mathbf{q}$ via:

$$\mathbf{q}^{t+1} = \mathcal{P}_{\Delta^n}(\mathbf{q}^t + \eta_{\mathbf{q}}^t \nabla_{\mathbf{q}} \mathcal{L}(\mathbf{w}^t, \mathbf{q}^t)). \quad (11)$$

Here $\mathcal{L}(\mathbf{w}, \mathbf{q}) = \sum_{e=1}^n q_e \mathcal{R}_{P_e}(\mathbf{w}) - \lambda \mathcal{D}(\mathbf{q}\|\mathbf{p})$, $\eta_{\mathbf{w}}$ ($\eta_{\mathbf{q}}$) are step sizes, and $\mathcal{P}_{\Delta^m}$ ($\mathcal{P}_{\Delta^n}$) denote projection (Duchi et al., 2008) onto the probability simplex.

By assigning higher weights to distributions with both high empirical risk and high centrality, our approach bridges the gap between overly conservative worst-case approaches and the fragility of average risk minimization under distribution shifts.

**Implementation Details**. (1) *Distribution distance $D(\cdot)$*. In contrast to traditional graph learning methods (Jin et al., 2020; Dong et al., 2019) which primarily focus on *point-wise* similarities, our approach models correlation on a *distribution-wise* level. While the Wasserstein distance is a natural choice for measuring distributional similarities, its standard computation has *cubic* complexity, posing practical challenges in large-scale applications. To address this, we adopt a computationally efficient Gaussian-based approximation of the 2-Wasserstein distance. For two Gaussian distributions $P_i = \mathcal{N}(\mu_i, \Sigma_i)$ and $P_j = \mathcal{N}(\mu_j, \Sigma_j)$, the 2-Wasserstein distance is: $W_2^2(P_i, P_j) = \|\mu_i - \mu_j\|_2^2 + \text{tr}(\Sigma_i + \Sigma_j - 2(\Sigma_j^{1/2} \Sigma_i \Sigma_j^{1/2})^{1/2})$. When the covariance matrices $\Sigma_i$ and $\Sigma_j$ commute, this simplifies to: $D^2(P_i, P_j) = \|\mu_i - \mu_j\|_2^2 + \|\Sigma_i^{1/2} - \Sigma_j^{1/2}\|_F^2$. This approximation maintains the key geometric properties of the Wasserstein distance while reducing computational complexity from cubic to quadratic time.

**Algorithm 1** Structure-informed Risk Minimization (SRM)

**Input:** Data of $\mathcal{E}_{\text{train}}$, Step sizes $\eta_{\mathbf{w}}$ and $\eta_{\mathbf{q}}$
**Output:** Learned ensemble weights $\mathbf{w}$
// Construct graph $\mathcal{G}$ and compute prior $\mathbf{p}$
  **for** $i, j \in \{1, \ldots, n\}$ **do**
    $D(P_i, P_j) \leftarrow \|\mu_i - \mu_j\|_2^2 + \|\Sigma_i^{1/2} - \Sigma_j^{1/2}\|_F^2$
    $A_{ij} \leftarrow D(P_i, P_j)$
**end**
  $c(P_e) \leftarrow [\sum_{j=1}^n d(P_e, P_j)]^{-1}$ // Closeness centrality
  $p_e \leftarrow c(P_e)/\sum_{j=1}^n c(P_j)$ // Prior distribution
  // Optimize weights
  Initialize $\mathbf{w}^0 \leftarrow \frac{1}{m}\mathbf{1}, \mathbf{q}^0 \leftarrow \frac{1}{n}\mathbf{1}$
  **while** *not converged* **do**
    Calculate $\mathcal{L}(\mathbf{w}, \mathbf{q})$ via Eq. 9
    Update ensemble weights $\mathbf{w}^{t+1}$ via Eq. 10
    Update mixture weights $\mathbf{q}^{t+1}$ via Eq. 11
**end**

(2) *Centrality prior* $\mathbf{p}$. We employ closeness centrality to compute the prior as it effectively captures a distribution's global influence across the entire graph, rather than just considering immediate neighbors. Specifically, for each distribution $P_e$, its closeness centrality is computed as: $c(P_e) = \left[\sum_{v \in \mathcal{V}} d(P_e, v)\right]^{-1}$, where $d(P_e, v)$ denotes the shortest path distance in the $\mathcal{G}$. The prior is then normalized as $p_e$ (Eq. 6). This formulation aligns well with our goal of identifying distributions that are most representative of the overall graph, as distributions with high closeness centrality are likely to share characteristics with a broader range of unseen test distributions, making them particularly informative for constructing robust uncertainty sets.

**Discussion**. (1) *Connection between $\tau$ (Eq. 7) and $\rho$ (Eq. 5)*. While DRO considers the uncertainty set defined as a divergence ball with a radius of $\rho$, our method operates on mixtures of training distributions. These formulations are closely connected: when the mixture weights $q$ satisfy $\mathcal{D}(\mathbf{q}\|\mathbf{p}) \leq \tau$, the resulting mixture distribution $P_q = \sum_{e=1}^n q_e P_e$ lies within a divergence ball centered at the reference distribution $P_p = \sum_{e=1}^n p_e P_e$. This provides a computationally tractable way to approximate divergence ball constraints while leveraging the structure induced by the distributional graph via centrality prior $\mathbf{p}$.

(2) *Connection to existing methods*. SRM provides a unified framework that encompasses existing approaches as special cases through the constraint $\mathcal{D}(\mathbf{q}\|\mathbf{p}) \leq \tau$. (i) When $\tau = 0$ and $p_e = \frac{1}{n}$, the constraint forces $\mathbf{q} = \mathbf{p}$, reducing to standard ERM that equally weights all training distributions. (ii) With $\tau = 0$ and centrality prior $\mathbf{p}$, we recover weighted risk minimization that prioritizes influential distributions. (iii) When $\tau = \infty$, the constraint becomes inactive, allowing $\mathbf{q}$ to be arbitrary, which recovers Group DRO (Sagawa et al., 2019) that optimizes for the worst-case distribution. By set-

ting $\tau \in (0, \infty)$, SRM interpolates between these extremes, leveraging the distributional graph to achieve a balance between average performance and worst-case robustness.

## 4. Experiments

We evaluate SRM on two common OoD generalization benchmarks, *DomainBed* (Gulrajani & Lopez-Paz, 2020) and *WILDS* (Koh et al., 2021). Following the standard practice, we use a held-out validation set from training distributions on *DomainBed* benchmark and validation distributions on *WILDS* benchmark for model selection. We provide implementation details and additional results in the Appendix. We provide the source code in the supplementary material.

**Baselines.** We compare SRM with the following methods: (1) Uniform Ensemble; (2) Greedy Selection; (3) Empirical Risk Minimization (ERM) (Vapnik & Vapnik, 1998); (4) Uniform Prior; (5) Laplacian Prior; (6) Group Distributionally Robust Optimization (DRO) (Sagawa et al., 2019). These methods can be grouped into two categories: (1) Non-optimization-based, where the ensemble weight is obtained without the need for optimization (Uniform Ensemble and Greedy Selection); (2) Optimization-based, where the ensemble weight is learned through an optimization process (ERM, Uniform Prior, Laplacian Prior and DRO).

### 4.1. DomainBed Benchmark

**Datasets**. We conduct experiments on five datasets: *TerraIncognita* (Beery et al., 2018), *VLCS* (Fang et al., 2013), *OfficeHome* (Venkateswara et al., 2017), *PACS* (Li et al., 2017), and *DomainNet* (Peng et al., 2019). *PACS* consists of images from four different distributions: art, cartoons, photos, and sketches. It contains a total of 9,991 images with dimensions of (3, 224, 224) pixels and 7 classes. *VLCS* contains photographic images from four distributions: Caltech101, LabelMe, SUN09, and VOC2007. There are 10,729 total images with dimensions of (3, 224, 224) pixels across 5 classes. *OfficeHome* is made up of images from four distributions: art, clipart, product images, and real-world photos. There are 15,588 images in this dataset with dimensions of (3, 224, 224) pixels and 65 classes. *TerraIncognita* consists of photos of wild animals captured by camera traps at four different locations. The dataset contains 24,788 images of size (3, 224, 224) pixels from 10 different classes. *DomainNet* is a large-scale dataset with images from six distributions: clipart, infographics, paintings, quickdraw sketches, real-world photos, and sketches. In total, there are 586,575 images of dimension (3, 224, 224) pixels across 345 classes. For each dataset, we hold one distribution out for test and train on the remaining ones, and report the average accuracies over all test distributions. Following (Gulrajani & Lopez-Paz, 2020), all the experimental results are averaged over 3 trials.

| Dataset | Non-optimization-based | | Optimization-based | | | | |
|---|---|---|---|---|---|---|---|
| | Uniform | Greedy | ERM | Uniform Prior | Laplacian | DRO | SRM |
| PACS | **87.8** | 87.1 | 86.9 | 87.0 | 87.0 | 87.1 | 87.1 |
| VLCS | 79.5 | 79.6 | 79.6 | 79.8 | 78.7 | 79.8 | **79.9** |
| OfficeHome | 70.8 | 70.8 | 70.7 | 70.7 | 70.8 | 70.5 | **71.0** |
| DomainNet | 43.3 | 44.6 | 45.0 | 45.0 | 45.0 | 45.0 | **45.1** |
| TerraIncognita | 48.5 | 48.1 | 48.8 | 48.8 | 48.8 | 49.0 | **49.5** |
| Average | 66.0 | 66.0 | 66.2 | 66.2 | 66.1 | 66.2 | **66.5** |

*Table 1.* Average accuracies (%) over all test distributions across datasets on *DomainBed* benchmark. We **bold** the best results.

| Test Distributions | Non-optimization-based | | Optimization-based | | | | |
|---|---|---|---|---|---|---|---|
| | Uniform | Greedy | ERM | Uniform Prior | Laplacian | DRO | SRM |
| Before 2004 | 53.7 | 53.3 | 54.4 | 54.1 | **54.4** | 53.2 | **54.4** |
| 2009-2011 | 62.3 | 62.3 | 62.3 | 62.4 | 62.3 | 62.3 | **62.5** |
| After 2016 | 37.6 | 37.5 | 37.7 | 37.7 | 37.8 | 37.0 | **37.8** |
| Average | 51.2 | 51.0 | 51.4 | 51.4 | 51.5 | 50.8 | **51.6** |

*Table 2.* Worst-region accuracy (%) on the FMoW-WILDS dataset. We evaluate the algorithms under three different train-test split schemes. SRM consistently outperforms other baselines in both *Distribution Interpolation* and *Distribution Extrapolation* settings.

**Results**. We report the results on *DomainBed* in Tab. 1. SRM achieves state-of-the-art performance on four datasets (*VLCS*, *OfficeHome*, *DomainNet*, and *TerraIncognita*), outperforming both non-optimization-based and optimization-based baselines. Notably, while other optimization-based methods deliver inconsistent results (*e.g.*, DRO outperforms ERM and Uniform Prior on *PACS* and *TerraIncognita*, but inferior to them on *OfficeHome*), SRM exhibits consistent improvements over prior optimization-based methods, with particularly significant gains on *TerraIncognita* (49.54% vs DRO's 48.96%). This indicates SRM achieves improved generalization across diverse distributions by taking distributional relations into account.

### 4.2. WILDS Benchmark

**Dataset.** We evaluate SRM on FMoW-WILDS (Koh et al., 2021) dataset, which comprises satellite images collected from different geographical regions across five continents at different time. We study temporal distribution shift, where distribution $d$ represents the year the image was taken. Apart from the original train-test split scheme (**Test After 2016**), where training distributions consist of years 2002 to 2013, test distributions consists of years 2016 and 2017, and years 2013 to 2016 are reserved for validation, we further propose two train-test split schemes which cover more diverse distribution shift scenarios: (1) **Test Before 2004**, where years 2007 to 2018 are for training, 2002 to 2004 are for testing, 2004 to 2007 are for validation; (2) **Test Middle**, where years 2002 to 2008 and years 2012 to 2018 are for training, 2009-2011 are for testing, 2008 and 2011 are for validation. These three settings cover both *Distribution Interpolation*

and *Distribution Extrapolation* cases proposed by (Wang et al., 2020), providing more thorough comparison of baselines than single train-test split scheme.

**Results.** Tab. 2 presents the worst-region accuracy for different methods. We observe that SRM consistently outperforms existing approaches across all three train-test settings. Notably, SRM improves worst-region accuracy compared to DRO, demonstrating its ability to mitigate over-pessimism while maintaining robustness. In the Test Before 2004 setting, SRM achieves 54.44%, outperforming DRO (53.21%) and standard ERM-based methods. In the Test Middle (2009-2011) setting, SRM attains 62.46%, slightly outperforming DRO and achieving the best generalization. In the Test After 2016 setting, SRM improves worst-region accuracy to 37.81%, surpassing DRO (36.98%), highlighting its advantage in handling temporal distribution shifts.

**Graph Visualization.** Fig. 1 visualizes the learned distribution graphs under different train-test split settings. Unlike DRO, which focuses on distributions with the worst test accuracy (even if they are far from the training distributions), SRM assigns higher weights to influential distributions with high centrality in the training graph. This structure-aware weighting strategy improves worst-region accuracy while avoiding excessive pessimism.

**Robustness to Severe Distribution Shift.** To test SRM's robustness to severe distribution shift, we simulate different level of distribution shift by adding corruptions to test data with different severity (Hendrycks & Dietterich, 2019). We apply two types of corruptions: *Blur* and *Digital*. The results under the setting of *Test After 2016* are reported in Fig. 2. As

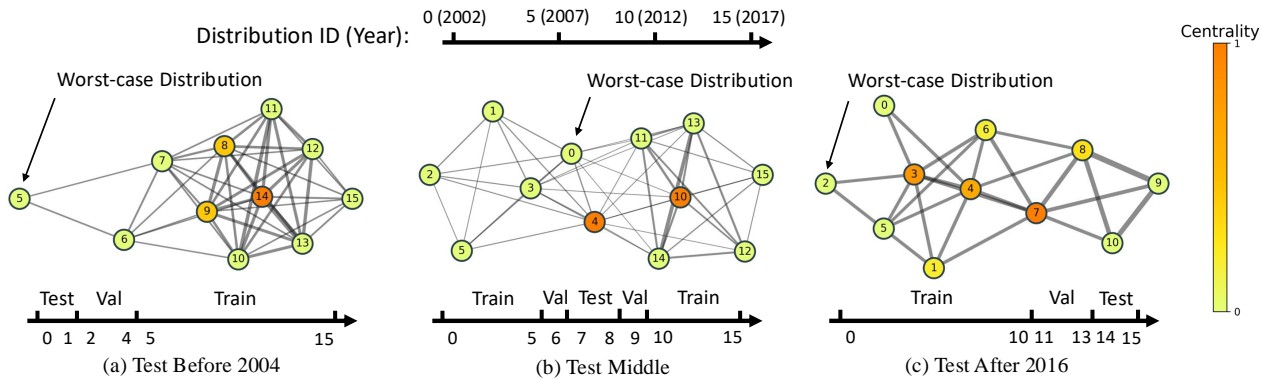

*Figure 1.* Visualization of distributional graph $\mathcal{G}$ under different train-test split schemes on FMoW-WILDS dataset. Distributions from year 2002 to year 2017 are labeled by 0 to 15, respectively. The thickness of edges indicates the similarity between distributions. While DRO focus on worst-case distributions (far from other distributions), SRM assigns higher mixture weights (**q**) to influential distributions.

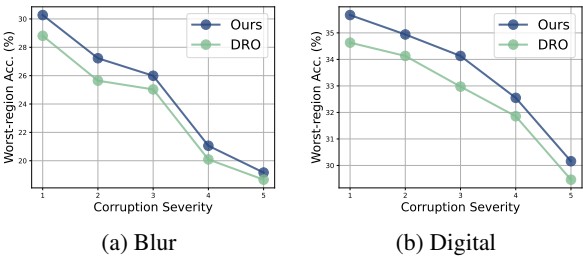

| (a) Blur | (b) Digital |
|---|---|

*Figure 2.* Worst-region accuracy (%) on FMoW-WILDS dataset under different severity of corruptions to test data. **Left:** results of *Blur* corruption. **Right:** results of *Digital* corruption. SRM consistently outperforms DRO under different types and different severity of corruptions.

observed, under different types and severity of corruptions, SRM consistently outperforms the DRO baseline, demonstrating the superior robustness to severe distribution shift of SRM. However, the performance gap between SRM and DRO is shrinking as the severity increases. This may attribute to SRM tries to construct a realistic uncertainty set while DRO focus on the worst-case distribution.

### 4.3. Ablation Study

We conduct the following ablation study under the *Test After 2016* setting to investigate how each component affects the effectiveness of SRM.

**Comparison with Individual Learners**. To further isolate the benefit of structure-informed weighting, we compare SRM with both individual models and a uniform ensemble baseline. Tab. 3 reports results on the DomainNet dataset across six test domains. SRM significantly outperforms both the average individual performance and uniform ensemble. For example, on the "Clipart" domain, accuracy improves from $56.79\%$ (mean of individual models) to $58.75\%$ (uniform ensemble) and then to $63.02\%$ with SRM, showing that leveraging structural relationships provides substantial

| Test Distribution | Individual Learners | | | | Ensemble | |
|---|---|---|---|---|---|---|
| | **min** | **max** | **mean** | **std** | **Uniform** | **SRM** |
| Clipart | 52.74 | 59.51 | 56.79 | 2.39 | 58.75 | 63.02 |
| Infograph | 18.19 | 20.12 | 18.98 | 0.61 | 21.30 | 21.77 |
| Painting | 44.99 | 48.36 | 46.57 | 1.09 | 51.07 | 51.92 |
| Quickdraw | 10.98 | 13.51 | 12.25 | 0.75 | 14.11 | 14.96 |
| Real-world | 57.31 | 61.30 | 59.67 | 1.09 | 62.20 | 64.51 |
| Sketch | 46.76 | 51.12 | 49.11 | 1.21 | 52.31 | 54.66 |

*Table 3.* Accuracy (%) of individual learners and ensemble methods on DomainNet dataset.

gains.

**Regularization Strength** $\lambda$. To assess the role of $\lambda$, which controls the trade-off between risk minimization and structural consistency, we vary $\lambda \in [0.0, 2.0]$ and report the worst-region accuracy in Fig. 3. When $\lambda = 0$, SRM reduces to Group DRO, which focuses purely on worst-case risk minimization without leveraging the structure of training distributions. As $\lambda$ increases, the worst-region accuracy improves, indicating that incorporating topological information helps refine the uncertainty set. However, overly large values of $\lambda$ (e.g., $\lambda > 1.5$) degrade performance, suggesting that excessive reliance on structural consistency may overly restrict the uncertainty set. Overall, a moderate $\lambda$ value (around 1.0) provides the best trade-off between robustness and average-case performance.

**Number of Models in the Ensemble Pool**. We analyze the effect of varying the number of models in the ensemble pool from 2 to 10. Fig. 4 presents the worst-region accuracy across different train-test splits. More models generally improve performance, as a larger ensemble provides greater flexibility. However, diminishing returns are observed beyond 7-8 models, indicating that excessive models may introduce redundancy.

**Distance Metric for Distribution Graph** $\mathcal{G}$. We evaluate

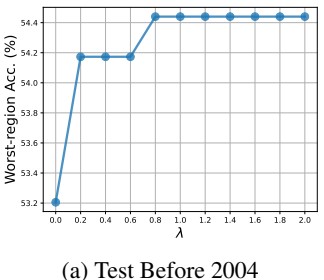
(a) Test Before 2004

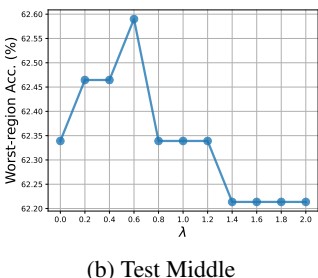
(b) Test Middle

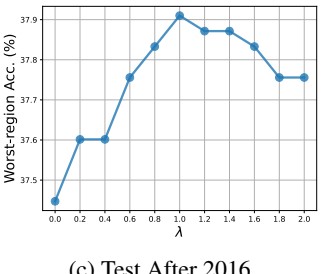
(c) Test After 2016

*Figure 3.* Ablation study of the regularization strength $\lambda$ (Eq. 9) on FMoW-WILDS dataset. Note that when $\lambda = 0$, SRM reduces to Group DRO. The worst-region accuracy first increases then decreases as $\lambda$ gets larger in *Test Middle* and *Test After 2016*. In *Test Before 2004*, the performance does not degrade because the worst-case distribution is far from other distributions (Fig. 1(a)), thereby leveraging structural prior consistently brings performance gain.

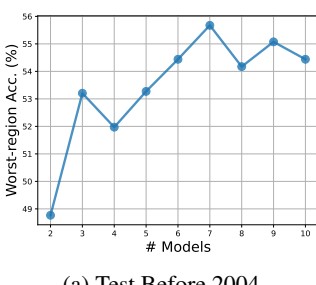
(a) Test Before 2004

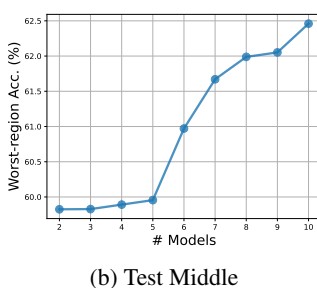
(b) Test Middle

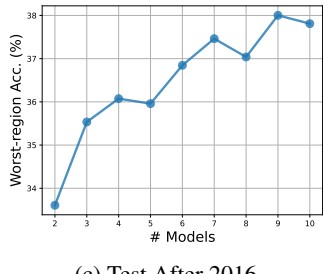
(c) Test After 2016

*Figure 4.* Ablation study of number of models ($m$) in the ensemble pool on FMoW-WILDS dataset. More models generally improves performance, while in some cases (*e.g.*, *Test Before 2004*) excessive models reduce efficiency due to model redundancy.

| Distance | Worst-region Acc. (%) | Running Time (s) |
|---|---|---|
| 2-Wasserstein | 38.1 | 28.1 |
| Diffusion EMD | 38.0 | 2.6 |
| EMD | 38.1 | 516.4 |

*Table 4.* Ablation study of distributional distance metrics for distribution graph $\mathcal{G}$. Our choice of 2-Wasserstein distance strikes the best balance between performance and time complexity.

three different metrics for computing pairwise distributional distances in the training graph $\mathcal{G}$: 2-Wasserstein Distance, Diffusion EMD (Diffusion Earth Mover's Distance (Tong et al., 2021)), and EMD (Standard Earth Mover's Distance). Tab. 4 reports worst-region accuracy for each method. Considering both performance and computational complexity, we choose 2-Wasserstein Distance for constructing the distribution graph.

**Comparison of Different Centrality Metrics**. To assess the role of graph centrality in defining the uncertainty set, we compare different centrality measures in Fig. 5. Closeness centrality achieves the highest worst-region accuracy, reinforcing our choice in the main method. Laplacian centrality and Katz centrality perform slightly worse, suggesting that they are less effective in capturing global influence. Betweenness centrality underperforms, likely due to its focus

on shortest paths rather than overall structure. Despite these, the performance gap between best and worst centrality is not obvious (0.07%), suggesting SRM is not sensitive to the choice of centrality.

**Graph Sparsity**. In our experiments, we use complete graph by default to calculate centrality. However, we also investigate how the graph sparsity (*i.e.* the percentage of edges retained in the distribution graph with respect to complete graph) influences performance by pruning edges between distant distributions. Fig. 6 shows that moderate sparsity (40-70 % edges) yields the best results. Excessive sparsity (below 30 % edges) degrades accuracy as most structural information is missing. These results suggest that balancing connectivity is crucial, as overly dense or sparse graphs lead to suboptimal uncertainty sets.

## 5. Related Work

Ensemble learning has emerged as a powerful paradigm for enhancing model robustness and improving out-of-distribution performance (Pagliardini et al., 2023; Lee et al., 2023). The fundamental principle underlying ensemble effectiveness lies in the strategic combination of diverse predictors, where diversity serves as a crucial factor in reducing ensemble prediction error (Ueda & Nakano, 1996). This diversity can manifest in various forms, including architectural

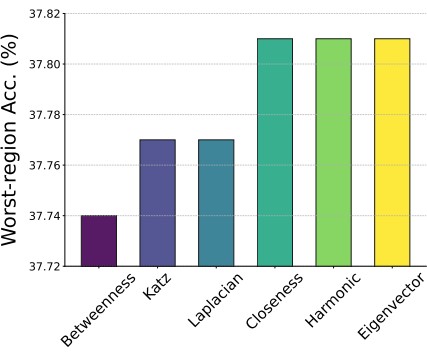

*Figure 5.* Ablation study of graph centrality. Performance gap between different choices of centrality is negligible.

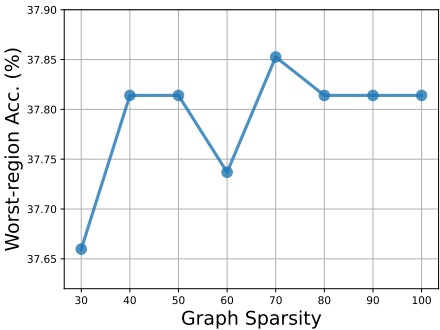

*Figure 6.* Ablation study of graph sparsity. Note "30" denotes 70% edges are pruned from the complete graph.

differences, training procedures, or learned representations that capture complementary aspects of the underlying data distribution.

Contemporary approaches to ensemble diversity optimization can be broadly categorized into two main strategies. The first category encompasses methods that directly optimize prediction diversity on target data (Pagliardini et al., 2023; Lee et al., 2023). These approaches explicitly encourage disagreement among ensemble members on specific data points, thereby promoting complementary decision boundaries. However, such methods inherently require access to test data during the optimization process, which presents a significant limitation in practical deployment scenarios where test distributions remain unknown at training time. The second category promotes diversity through the adoption of varied learning procedures (Arpit et al., 2022; Rame et al., 2022; Wortsman et al., 2022). These methods generate ensemble diversity by systematically varying training configurations, including hyperparameter settings, optimization algorithms, data augmentation strategies, initialization schemes, and architectural choices. While this approach offers the advantage of not requiring test data access, it faces the fundamental challenge that diverse training procedures

do not guarantee diverse predictive behaviors. Consequently, such methods may inadvertently generate redundant models that exhibit similar decision boundaries, potentially compromising ensemble performance when combined through uniform weighting schemes.

The challenge of effective model selection within ensemble frameworks represents another critical consideration. A prevalent practice in the field is greedy selection (Wortsman et al., 2022), where individual models are sequentially incorporated into the ensemble based on their contribution to validation accuracy improvements. This validation-centric approach, while computationally efficient and theoretically motivated, suffers from a fundamental limitation: validation-based selection criteria may not adequately generalize to shifted test distributions that differ significantly from the validation set. This generalization gap becomes particularly pronounced in scenarios involving substantial domain shifts or distributional changes.

Distributionally Robust Optimization (DRO) (Shalev-Shwartz & Wexler, 2016) provides a principled mathematical framework for addressing distribution shifts by optimizing worst-case performance over a carefully constructed set of potential test distributions. Traditional DRO formulations define uncertainty sets using various divergence metrics, including $f$-divergence (Namkoong & Duchi, 2016), Wasserstein distance (Shafieezadeh Abadeh et al., 2018; Qiao et al., 2020), and other statistical distance measures. These uncertainty sets characterize the space of plausible distribution shifts that the model might encounter during deployment.

Despite the theoretical elegance and empirical success of DRO in training individual robust models, the framework suffers from a well-documented over-pessimistic issue. This over-pessimism manifests as excessive conservatism in the worst-case optimization objective, often leading to suboptimal performance on the actual test distribution. Several methods (Liu et al., 2022; Qiao & Peng, 2023; Ma et al., 2024; Huang & Ding, 2025) have been proposed to mitigate this over-pessimistic behavior. However, these existing solutions predominantly focus on training individual models rather than learning robust ensemble combinations, leaving a significant gap in the ensemble learning literature.

## 6. Conclusion

We proposed Structure-informed Risk Minimization (SRM), a principled framework for OoD generalization in ensemble learning. The innovation lies in utilizing distributional graphs to construct uncertainty sets that focus on plausible distribution shifts. By incorporating the relationships between training distributions, SRM achieves a balance between robustness to unseen shifts and strong average-case performance. Our theoretical analysis establishes guaran-

tees for both convergence and generalization, while extensive experiments across diverse benchmarks demonstrate SRM's effectiveness in real-world scenarios.

## Acknowledgments

This work is supported by the National Science Foundation under grant numbers CAREER 2340074, SLES 2416937, III CORE 2412675, National Institutes of Health under grant number R21CA301093, Department of Defense under grant number AFOSR FA9550-23-1-0494.

## Impact Statement

This paper presents work whose goal is to advance the field of Machine Learning. There are many potential societal consequences of our work, none which we feel must be specifically highlighted here.

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

# A. Experiments

## A.1. Implementation Details

We use DiWA (Rame et al., 2022) to train the models in the ensemble pool. Each model in the ensemble pool is a ResNet-50 (He et al., 2016) model trained with ERM (Vapnik & Vapnik, 1998) using different hyper-parameter settings. The number of models ($n$) used in the experiments is 10. A random model in the ensemble pool is chosen to construct the distribution graph. For optimizing $w$ and $q$, we use SGD optimizer. For the experiments on *DomainBed*, we set $\eta_w = 0.1$ and $\eta_q = 0.1$, and for *WILDS*, we set $\eta_w = 3e^{-2}$ and $\eta_q = 0.1$. $\lambda$ is selected from $[0.0, 2.0]$ for each dataset. We use in-distribution validation set to optimize $w$ and $q$, and the number of steps is 100 and 50 for *DomainBed* and *WILDS*, respectively.

## A.2. Additional Results on DomainBed

We provide detailed experiment results for each test environment on *DomainBed* benchmark.

| Env. | Non-optimization-based | | Optimization-based | | | | |
|------|---------|--------|------|---------------|-----------|------|------|
|      | Uniform | Greedy | ERM  | Uniform Prior | Laplacian | DRO  | SRM  |
| **PACS** | | | | | | | |
| Art | 88.2 | **88.5** | 88.0 | 88.3 | 88.0 | 88.0 | 88.5 |
| Cartoon | **82.1** | 80.0 | 80.3 | 80.2 | 80.3 | 80.4 | 80.4 |
| Photo | **98.5** | 98.3 | 98.0 | 98.0 | 98.0 | 98.1 | 98.1 |
| Sketch | **82.6** | 81.6 | 81.5 | 81.3 | 81.5 | 81.8 | 81.8 |
| **VLCS** | | | | | | | |
| Caltech101 | 98.2 | 98.8 | 99.1 | 99.2 | 98.5 | 99.1 | **99.3** |
| LabelMe | 64.5 | 64.3 | 65.4 | 65.3 | 65.1 | 65.3 | **65.4** |
| SUN09 | 75.3 | **75.6** | 73.9 | 74.3 | 72.4 | 74.4 | **75.6** |
| VOC2007 | 80.1 | 79.8 | 80.3 | 80.5 | 78.7 | 80.3 | **80.6** |
| **OfficeHome** | | | | | | | |
| Art | **67.6** | 67.3 | 66.9 | 66.8 | 66.7 | 66.7 | 67.3 |
| Clipart | 56.9 | **57.0** | 56.4 | 56.3 | 56.3 | 56.1 | **57.0** |
| Product | 78.1 | 78.3 | 79.4 | 79.2 | 79.5 | 79.1 | **79.6** |
| Real-world | 80.6 | 80.6 | 80.4 | 80.4 | **80.6** | 80.2 | **80.6** |
| **DomainNet** | | | | | | | |
| Clipart | 58.8 | 62.0 | 63.0 | 62.9 | 62.9 | 62.9 | **63.0** |
| Infograph | 21.3 | 21.6 | 21.7 | 21.7 | 21.7 | 21.7 | **21.8** |
| Painting | 51.1 | 51.7 | 51.6 | 51.6 | 51.5 | 51.6 | **51.9** |
| Quickdraw | 14.1 | 15.5 | 15.0 | 14.9 | 14.9 | 14.9 | **15.0** |
| Real-world | 62.2 | 63.3 | 64.4 | 64.4 | 64.4 | 64.4 | **64.5** |
| Sketch | 52.3 | 53.4 | 54.5 | 54.4 | 54.6 | 54.4 | **54.7** |
| **TerraIncognita** | | | | | | | |
| L100 | 55.3 | **58.9** | 55.1 | 55.8 | 55.8 | 56.2 | **58.9** |
| L38 | 40.8 | 39.6 | 40.9 | 40.6 | 40.7 | 40.9 | **41.5** |
| L43 | 60.4 | 60.6 | 60.8 | 60.7 | 60.6 | 60.9 | **61.1** |
| L46 | 37.3 | 33.4 | 38.4 | 38.1 | 38.2 | 37.9 | **38.9** |

*Table 5.* Average accuracy (%) over all test distributions for PACS, VLCS, OfficeHome, DomainNet, and TerraIncognita datasets.

