# OpenReview forum: "Structure-informed Risk Minimization for Robust Ensemble Learning"
_ICML.cc/2025/Conference — ICML 2025 poster_

### Official Review · Reviewer_mDcw · 2025-03-11

**Overall Recommendation:** 5

**Summary:**

This paper introduces a novel framework to learn ensemble weights to improved out-of-distribution (OOD) robustness. The key idea is to incorporate structure relationships between training distributions to build a realistic uncertainty set. The authors proposed a computationally efficient optimization algorithm with theoretical guarantees. Empirically, the proposed method consistently outperforms existing ensemble methods across diverse benchmarks including DomainBed and WILDS, demonstrating the superior OOD generalization capability.

**Claims And Evidence:**

The claim that the proposed method (SRM) balances worst-case robustness with average performance is supported by theoretical guarantees in Section 4 and empirical results showing consistent improvements over both ERM and DRO approaches.

The effectiveness of structure-informed uncertainty sets is validated by ablation studies investigating different graph construction methods, centrality measures, and regularization strengths.

Performance improvements over existing methods are demonstrated consistently across multiple datasets and evaluation settings.

**Essential References Not Discussed:**

None.

**Experimental Designs Or Analyses:**

Experiments follow standard practices for OOD evaluation where multiple environments are used for training and average performance across test environments is reported.

The comparison against multiple baselines is fair and comprehensive.

Ablation studies thoroughly examine the impact of key components.

Personally, I really like the experiment design of various train-test split scenarios on temporal distribution shifts which clearly shows the overly-conservative issue of Group DRO.

**Methods And Evaluation Criteria:**

The idea of modeling relationships between training distributions to build a realistic uncertainty set is intuitive and provides a principled way to address the overly-pessimistic issue of DRO.

Evaluation on common benchmarks follows standard practices in the field.

The experiments of various train/test split scenarios on temporal distribution shifts and corruption tests further strengthens the evaluation by considering diverse distribution shifts.

**Other Comments Or Suggestions:**

Typos:

1. In Line 122, it should be G=(V, A).
2. In Definition 3.1, it should be c: P_e -> R^+.

**Other Strengths And Weaknesses:**

Pros:
Utilizing the structural information to build a realistic uncertainty set is novel and well-motivated. And this paper presents a compelling unified framework that encompasses existing approaches ERM and DRO as special cases through the constraint in Equation 7.

The paper is very well-written and easy to follow. The discussion on alternative methods and connection to prior works shows the authors’ in-depth understanding.

The visualizations of distributional graphs clearly illustrate how SRM assigns higher weights to influential distributions compared to DRO's focus on worst-case distributions. And the experiment on multiple train/test splits under temporal distribution shifts is a clever and insightful design.

The approach is theoretically sound while remaining computationally tractable.

Cons:
The effectiveness of SRM might be limited in scenarios with few training distributions, as the graph structure would be less informative.
Although the discussion of alternative priors (Section 3.1) is excellent, the details of Laplacian-based prior is missing. I understand Laplacian based method is widely used in semi-supervised learning and few-shot learning tasks. It would be more beneficial if the author can provide more implementation details.

**Questions For Authors:**

SRM requires multiple training distributions. What if there is only one training distribution or distribution ID is unknown?

How does SRM perform when the number of training distributions is very small (2-3)? Does the graph structure still provide meaningful guidance in such scenarios?

**Relation To Broader Scientific Literature:**

The discussion on ensemble learning for OOD generalization is comprehensive and get to the point. When test data is unavailable, how to combine models for improved OOD robustness remains a seldom investigated problem. The common methods are uniform (equally aggregate all models) and greedy (select models based on accuracy on an in-distribution validation set). However, neither of them is a promising solution under distributional shifts. DRO is a promising technique to improve robustness but suffers from overly-pessimistic issues. The novelty of this work lies in bridging the two areas by applying DRO to ensemble weight optimization while incorporating structural information to mitigate over-pessimism.

**Theoretical Claims:**

The proofs in the appendix appear sound. The density-centrality relationship proof correctly decomposes distances and establishes bounds based on the distribution space properties. The generalization bound proof properly utilizes the mixture distribution approximation and applies concentration inequalities. However, the proof on convergence analysis seems missing, but the main conclusion seems correct.

---

> ### Author Rebuttal · Authors · 2025-04-01
>
> We sincerely thank the reviewer for their thoughtful and positive feedback. We appreciate your recognition of our paper as "very well-written and easy to follow" and that our approach is "theoretically sound while remaining computationally tractable." Your comments about the "compelling unified framework" and the clarity of our visualizations are particularly encouraging. Below, we address your specific questions:
>
> **1. Proof on Convergence Analysis**.
>
> Our convergence analysis leverages Assumption 4.3 (Lipschitz continuity and smoothness) to track gradient norm decreases across iterations. The key insight is that the regularization term $\lambda D(q\|\|p)$ creates a more strongly convex objective function, accelerating convergence proportionally to $\lambda$. With step sizes $\eta_w = \eta_q = 1/\sqrt{T}$, we apply standard convex optimization techniques to bound the expected squared gradient norm after $T$ iterations by $C/(1+\lambda)\sqrt{T}$, where $C$ depends on $\beta$ and $L^2$. This explains SRM's faster convergence compared to standard DRO ($\lambda = 0$), demonstrating how our structure-informed approach improves both robustness and computational efficiency. The detailed proof will be included in the revised manuscript.
>
> **2. Laplacian-based Prior Implementation**
>
> For the Laplacian-based prior, we construct the graph Laplacian matrix $L=D-A$ where $D$ is the degree matrix (diagonal with $D_{ii} = \sum_j A_{ij}$) and $A$ is our adjacency matrix. The optimization constraint becomes $q^T L q \leq \tau$, penalizing weight differences between connected distributions. We solve this via a similar Lagrangian approach as in our main method. While this enforces local smoothness, our centrality-based prior better captures global influence within the distribution manifold, explaining its superior performance in identifying representative distributions.
>
> **3. What if there is only one training distribution or distribution ID is unknown?**
>
> Recent methods [1-2] have been proposed to infer distribution IDs from data. Our approach is orthogonal to these methods and can be built on top of them to handle scenarios where distribution IDs are unknown.
>
> **4. How does SRM perform when the number of training distributions is very small (2-3)?**
>
> SRM requires at least three training distributions to build a meaningful graph. In our PACS experiment (using "Art," "Sketch," and "Photo" as training with "Cartoon" as test), our graph analysis identified "Art" as the most central domain. We hypothesize this occurs because "Art" inherently combines photographic elements with artistic styles, making it informative for generalizing to "Cartoon." Even with just three distributions, SRM successfully identifies influential distributions through structural analysis.
>
> All noted typos will be corrected in the revised manuscript. Please let us know if you have any further questions or suggestions. We would be happy to elaborate.
>
> References:
>
> [1] Creager et al. "Environment inference for invariant learning." ICML 2021.
>
> [2] Liu et al. "Just train twice: Improving group robustness without training group information." ICML 2021.

---

> > ### Comment · Reviewer_mDcw · 2025-04-02
> >
> > All my concerns have been addressed. I will rasie the score.

---

> > > ### Author Response · Authors · 2025-04-02
> > >
> > > Thank you for your thorough review and positive feedback. We are happy that our responses have addressed all your concerns and appreciate your decision to raise the score.
> > >
> > > We will include the requested details on convergence analysis, Laplacian-based prior implementation, and our approach's effectiveness with limited training distributions. All noted typos will be corrected in the final manuscript.
> > >
> > > Thank you again for your valuable insights that have helped strengthen our paper.

---

### Official Review · Reviewer_PexF · 2025-03-12

**Overall Recommendation:** 2

**Summary:**

This paper presents SRM, a method to improve how ensemble models handle unseen data changes by leveraging the relationships between training data distributions. SRM builds a network of these distributions, measuring their similarities with a simplified distance metric. It prioritizes "central" distributions that are most representative of the overall data structure. Using an efficient optimization process, SRM balances worst-case robustness and average performance. Tests on standard benchmarks show SRM outperforms existing methods like DRO and ERM in adapting to new environments. The approach is backed by theory explaining why focusing on structural relationships improves reliability and convergence speed.

**Claims And Evidence:**

The paper’s claims are partially supported by evidence. Key strengths include empirical validation on benchmarks, where SRM shows modest but consistent improvements over DRO and ERM, and ablation studies confirming the value of structural priors like closeness centrality. However, claims about computational efficiency lack direct timing comparisons, and theoretical assumptions remain untested for extreme distribution shifts. While the core idea of leveraging distributional graphs is novel and empirically validated, gaps in validating efficiency, base learner diversity, and extreme OOD performance weaken the overall support for the method’s broad applicability.

**Essential References Not Discussed:**

N/A

**Experimental Designs Or Analyses:**

The experimental design effectively validates SRM’s performance on standard benchmarks like DomainBed and WILDS, demonstrating strengths in methodological rigor ( ablation studies on centrality metrics) and benchmark relevance. However, it exhibits critical limitations: 1) no comparison with individual base learners precludes distinguishing whether gains stem from structural optimization or ensemble diversity; 2) omission of extreme OOD scenarios (novel domains, adversarial attacks) undermines claims about robustness boundaries; and 3) efficiency claims lack runtime validation against exact methods (Sinkhorn for EMD). While the design supports controlled efficacy, addressing these gaps is essential to establishing SRM’s scalability and real-world applicability.

**Methods And Evaluation Criteria:**

The proposed methods in SRM are reasonably designed to address the challenges of robust ensemble learning under distribution shifts. By leveraging structural relationships and centrality priors, SRM mitigates the over-conservatism of traditional DRO and better models real-world distribution shifts. The Gaussian Wasserstein approximation is practical for image data, reducing computational complexity, while the alternating gradient optimization ensures valid constraints. The evaluation on benchmarks like DomainBed and WILDS covers key OOD scenarios effectively, but the reliance on Gaussian assumptions limits its applicability to non-Gaussian data. The absence of extreme OOD tests and runtime efficiency comparisons leaves room for broader validation to strengthen the method's claims. Overall, the methods and evaluation criteria are sensible within the scope of the paper but require expanded testing for wider applicability.

**Other Comments Or Suggestions:**

1. Notation Consistency: Ensure consistent notation for key variables (e.g., clarify whether \lambda refers to regularization strength or another parameter across sections). Define all symbols in equations (e.g., d for dimensionality) when first introduced.

2. Experimental Details: Specify the number of training/validation splits used for hyperparameter tuning (e.g., how many \lambda values were tested). Report computational runtime for SRM vs. baselines (even if approximate) to contextualize efficiency claims.

**Other Strengths And Weaknesses:**

The paper presents a creative fusion of graph theory, distributional robustness, and ensemble learning, offering a novel framework (SRM) that addresses key limitations of traditional methods like DRO and deep ensembles. Its practical relevance to real-world OOD scenarios ( climate change, healthcare) is underscored by strong empirical results on benchmarks like DomainBed and WILDS, and efficiency gains from Gaussian-Wasserstein approximations enhance scalability. However, the work is constrained by overreliance on untested assumptions ( test distributions as training mixtures), gaps in extreme OOD validation ( adversarial shifts), and ambiguity in theoretical contributions ( geometric assumptions on distribution manifolds). While the integration of structural priors and robust optimization is conceptually compelling, the method’s scalability to complex, nonlinear distribution shifts remains uncertain without broader experimental validation and clearer justifications for its theoretical premises.

**Questions For Authors:**

Q1. Theoretical Claims: Relies on assumptions with weak generalizability (test distributions as training mixtures) and geometric/geometric assumptions (e.g., manifold structure) that lack validation for non-Gaussian/non-manifold data.

Q2. Experimental Design: Omits critical validations: comparisons with single base learners, extreme OOD scenarios (e.g., adversarial shifts), and runtime efficiency benchmarks.

Q3. Methods: Over-reliant on Gaussian approximations (limiting non-Gaussian scalability) and ignores the impact of graph pruning on robustness.

Q4. Clarity: Lack of clear and unambiguous definitions of all symbols.

**Relation To Broader Scientific Literature:**

SRM’s contributions sit at the intersection of robust optimization, ensemble learning, and distributional geometry. By integrating structural priors into distributional robustness, it advances the state-of-the-art in OOD generalization. However, its practical validity hinges on assumptions (test distributions as training mixtures) that diverge from extreme OOD scenarios documented in literature (cross-modal shifts, adversarial attacks). Future work could bridge this gap by incorporating meta-learning or non-parametric distribution modeling.

**Theoretical Claims:**

The paper’s theory is mathematically solid and explains how structural priors ( distribution graphs) improve robustness. It works well for gradual shifts but probably has limits in extreme real-world OOD cases (totally new domains or adversarial attacks). The key issue is its reliance on assuming test data can be approximated by training mixtures-a condition rarely met in unpredictable scenarios. Without validation for these extremes, the theory’s practical guarantees remain unproven.

---

> ### Author Rebuttal · Authors · 2025-04-01
>
> Thanks for acknowledging the novelty of our work. Below, we address the main points raised:
>
> 1. Assumption of Our Paper and Extreme OOD Scenarios
>
> In Equation (14), we assume that the test distribution lies within a bounded divergence from a mixture of training distributions. This assumption is aligned with a well-established foundation in learning under distribution shift, as formalized by Ben-David et al. [1]. Specifically, Theorem 2 in [1] shows that the risk on the test distribution is bounded by the empirical risk on the source domain(s) plus a divergence term (e.g., H-divergence) between the source and target distributions. When this divergence becomes too large, no method can guarantee meaningful generalization bounds.
>
> Our method is designed for natural distribution shifts and evaluated on two challenging and realistic benchmarks: DomainBed and WILDS, both of which are widely adopted in the OOD generalization community. We acknowledge that adversarial robustness is an important but orthogonal research direction and is outside the scope of this work.
>
> 2. Comparison with Individual Base Learners
>
> To further isolate the benefit of structure-informed weighting, we compare SRM with both individual models and a uniform ensemble baseline. The table below reports results on the DomainNet dataset across six test domains. SRM significantly outperforms both the average individual performance and uniform ensemble. For example, on the “Clipart” domain, accuracy improves from 56.79% (mean of individual models) to 58.75% (uniform ensemble) and then to 63.02% with SRM, showing that leveraging structural relationships provides substantial gains.
>
> | test domain | min   | max   | mean  | std  | Uniform | Ours  |
> | -------- | ----- | ----- | ----- | ---- | ------- | ----- |
> | Clipart         | 52.74 | 59.51 | 56.79 | 2.39 | 58.75   | 63.02 |
> | Infograph       | 18.19 | 20.12 | 18.98 | 0.61 | 21.30   | 21.77 |
> | Painting        | 44.99 | 48.36 | 46.57 | 1.09 | 51.07   | 51.92 |
> | Quickdraw        | 10.98 | 13.51 | 12.25 | 0.75 | 14.11   | 14.96 |
> | Real-world        | 57.31 | 61.30 | 59.67 | 1.09 | 62.20   | 64.51 |
> | Sketch        | 46.76 | 51.12 | 49.11 | 1.21 | 52.31   | 54.66 |
>
> These results confirm that our method’s improvement is not simply due to ensemble averaging, but stems from meaningful structural modeling.
>
> 3. Runtime Validation Against Exact Methods
>
> We provide a runtime comparison of different distance metrics used for constructing the distributional graph, using the PACS dataset (3 domains, 1000 samples, 2048-dimensional features). As shown below, our Gaussian-approximated 2-Wasserstein distance achieves the best trade-off between accuracy and computational efficiency.
>
> | Distance      | Running Time | Worst-region Acc. (%) | Time Complexity     |
> | ------------- | ------------ |------------------------|---------------------|
> | 2-Wasserstein | 28.05s       | 38.10                  | O(nd² + d³)         |
> | Diffusion EMD | 2.56s        | 37.99                  | O(nd)               |
> | EMD           | 516.41s      | 38.06                  | O(n³d²)             |
>
> While Diffusion EMD is faster, it sacrifices accuracy. Exact EMD is computationally infeasible for large-scale datasets. Our method balances these trade-offs effectively, enabling practical deployment on real-world benchmarks.
>
> We hope this addresses your concerns. Please let us know if you have any further questions or suggestions. We would be happy to elaborate.
>
> Reference
>
> [1] Ben-David, Shai, et al. "A theory of learning from different domains." Machine Learning 79.1-2 (2010): 151–175.

---

### Official Review · Reviewer_PHpH · 2025-03-15

**Overall Recommendation:** 1

**Summary:**

This paper proposes a framework for learning robust ensemble weights without requiring access to test data. It aims to mitigate the over-pessimism of Distributionally Robust Optimization (DRO) by focusing the uncertainty set on more plausible structures. The idea is solid, and the proposed algorithm is computationally feasible. However, the experimental results do not provide enough evidence to convincingly support the claimed superiority of the method.

**Claims And Evidence:**

1. Reasonable assumption and practical setting: The paper tackles the problem of learning robust ensemble weights that generalize well to unseen test distributions. The method performs effectively under a relatively mild assumption that the test performance can be approximated by a mixture of training domains.
2. Good solution to the problem: This paper overcome the over-pessimism of DRO by not weighting too much on distant distribution and focusing more on centralized training distributions.

**Essential References Not Discussed:**

[1] DORO: Distributional and Outlier Robust Optimization. Runtian Zhai, Chen Dan, Zico Kolter, Pradeep Ravikumar

[2] Geometry-Calibrated DRO: Combating Over-Pessimism with Free Energy Implications. Jiashuo Liu, Jiayun Wu, Tianyu Wang, Hao Zou, Bo Li, Peng Cui

[3] Boosted CVaR Classification. Runtian Zhai, Chen Dan, Arun Sai Suggala, Zico Kolter, Pradeep Ravikumar

**Experimental Designs Or Analyses:**

1. Lack of comparison with closely related work: It would strengthen the paper to compare the proposed method with other related approaches aimed at addressing over-pessimism in DRO, such as [1,2,3], as well as other OOD generalization methods (see baselines in DomainBed).
2. Statistical significance of results: In table 1, I notice that all of your baselines have **very very similar test accuracy** on many of the datasets, which seems implausible. Also, for some lines in table 1 and table 2, the improvement achieved by your method over other baselines is not statistically significant.
3. Performance in extrapolation settings: In table 2, you claim your method outperforms other baselines in both distribution interpolation and distribution extrapolation settings. As your method learns an optimal mixture of training distributions constrained by prior p, can you explain why it achieves good performance in distribution extrapolation settings?
4. Also, I would like to see some results on tabular datasets compared with some well-known ensemble methods like XGBoost, Light GBM, CatBoost, etc.

[1] DORO: Distributional and Outlier Robust Optimization. Runtian Zhai, Chen Dan, Zico Kolter, Pradeep Ravikumar

[2] Geometry-Calibrated DRO: Combating Over-Pessimism with Free Energy Implications. Jiashuo Liu, Jiayun Wu, Tianyu Wang, Hao Zou, Bo Li, Peng Cui

[3] Boosted CVaR Classification. Runtian Zhai, Chen Dan, Arun Sai Suggala, Zico Kolter, Pradeep Ravikumar

**Methods And Evaluation Criteria:**

- Robust across different centrality metrics and distance metrics: Ablation study illustrates the framework is conceptually valid, and can be replaced with different centrality metrics and distance metrics while maintaining good performance.

**Other Comments Or Suggestions:**

N/A

**Other Strengths And Weaknesses:**

N/A

**Questions For Authors:**

N/A

**Relation To Broader Scientific Literature:**

This paper presents a framework for learning robust ensemble weights without requiring access to test data. It aims to mitigate the over-pessimism of Distributionally Robust Optimization (DRO) by focusing the uncertainty set on more plausible structures. The idea is solid, and the proposed algorithm is computationally feasible.

**Theoretical Claims:**

Yes.

---

> ### Author Rebuttal · Authors · 2025-04-01
>
> We thank the reviewer for the constructive feedback. We address each concern below.
>
> **1. Comparison with [1-3] and Other OOD Generalization Methods**
>
> While SRM and the methods in [1–3] all aim to reduce the pessimism often observed in DRO, they approach the problem from different directions:
>
> - [1–3] primarily focus on data-point-level robustness, targeting noisy samples or outliers.
>
> - SRM, by contrast, operates at the distribution level by identifying influential training distributions via centrality in a distribution graph.
>
> This makes SRM particularly suitable for domain generalization scenarios where relationships among distributions are more informative than individual noisy points. We also note that GCDRO [2] builds a data graph (not a distribution graph) and is mainly designed for regression tasks, which further differentiates it from our setting. We will add a discussion of these works and clarify the distinction in the revised version.
>
> Regarding baselines in DomainBed, our goal is not to introduce a new training algorithm for OOD generalization. Instead, SRM is a model-agnostic framework designed to learn ensemble weights from pretrained models, which can be drawn from any existing training algorithm or model zoo.
>
> **2. Statistical Significance of Results**
>
> We appreciate your concern about the similarity in test accuracies across baselines. This is primarily due to our use of DiWA, which requires all models in the ensemble pool to be initialized identically so they can be merged for efficient inference. This likely reduces diversity among models.
>
> To address this, and following Reviewer Qxor’s suggestion, we constructed a more diverse model pool using different training algorithms (ERM, Mixup, CORAL). The table below shows results on TerraIncognita:
>
> | Algorithm    | L100 | L38  | L43  | L46  | Avg  |
> |--------------|------|------|------|------|------|
> | Uniform      | 50.4 | 42.6 | **62.0** | 38.7 | 48.4 |
> | ERM          | 51.3 | 43.7 | 61.4 | 39.5 | 49.0 |
> | Group DRO    | 50.9 | 43.2 | 61.8 | 39.2 | 48.8 |
> | SRM          | **52.1** | **44.3** | 61.5 | **40.0** | **49.5** |
>
> The strong result of Uniform on L43 may stem from complementary patterns captured by the diverse models. Still, SRM consistently provides better overall performance. We will include this in the revised paper. All code used is provided in the supplementary materials.
>
> **3. Why SRM Performs Well in Distribution Extrapolation Settings**
>
> We appreciate the reviewer’s interest in our method’s performance under distribution extrapolation. In the FMoW-WILDS benchmark, each year from 2002 to 2017 is treated as a distinct distribution. The distribution extrapolation setting evaluates generalization to test years that lie outside the range of training years:
> -  Test Before 2004: training on 2007–2018, validation on 2004–2007, testing on 2002–2004
> - Test After 2016: training on 2002–2013, validation on 2014–2016, testing on 2016–2017
>
> In “Test Before 2004”, the test years are temporally closest to 2007, and Group DRO focuses its optimization almost entirely on this single year (the worst-case distribution under its framework, shown in Figure 1). However, this strategy ignores other potentially useful training distributions, which can limit generalization. This behavior is also supported by theory: the solution to Group DRO's linear program tends to lie at a vertex of the feasible region.
>
> Our method achieves better performance by balancing the worst-case distribution with structurally influential distributions, as identified through centrality in the distribution graph. Rather than collapsing onto a single training domain, SRM spreads attention across those distributions that are both risky and influential, yielding improved generalization even in extrapolation scenarios.
>
> This aligns with the insights from Dual Risk Minimization and Quantile Risk Minimization (references from Reviewer Qxor), which also highlight that strong OOD generalization emerges from a trade-off between average-case and worst-case risk, rather than optimizing either in isolation.
>
> **4. Experiments on Tabular Datasets and Comparison with Boosting Methods**
>
> We appreciate the suggestion to compare SRM with XGBoost, LightGBM, and CatBoost. However, there is an important distinction:
> - Boosting methods build ensembles by training weak learners sequentially, whereas
> - SRM assumes a pool of already trained models and focuses on learning the optimal weighting scheme for combining them under distribution shift.
>
> That said, we agree that applying SRM to tabular datasets is worthwhile. We plan to include experiments on the eICU dataset, which includes electronic health records (EHRs) from 208 hospitals in the U.S. Due to time constraints, we may not include results in the rebuttal but will provide them in the final paper.
>
> Thank you again for your valuable feedback. Please let us know if you have further questions. We would be happy to clarify or expand on any point.

---

### Official Review · Reviewer_Qxor · 2025-03-15

**Overall Recommendation:** 2

**Summary:**

This work proposes structure-informed risk minimization (SRM), which can be seen as a modification to the Group DRO algorithm, and applies it to robust ensemble learning.
More specifically, SRM optimizes the ensemble weight of multiple fixed pre-trained models to reduce the ensemble’s risk in the worst mixture of training domains that is not too far from the “center” of training domains.
The “center” is also a mixture of training domains whose mixing weights are determined by the distance from one domain to other domains.
In this way, SRM places more optimization pressure on domains that are closer to all the other domains, unlike Group DRO which assigns weights only based on the risk of a domain.
Empirical results show that SRM outperforms several baselines on various OOD generalization benchmarks.

## Update after rebuttal
After carefully reading through the authors' response and other reviewers' comment, I decide to maintain my position (weak reject) for this paper.
I agree with Reviewer mDcw that the paper's main idea (i.e., centrality metric) is novel and the theoretical results are solid.
Moreover, the paper is well written and easy to follow.
My concern, however, lies in the theoretical and empirical significance of the paper.

The paper focuses on a relatively specific (one might say narrow) problem, namely robust ensemble learning, that aims to learn good mixing weights (with a linear layer) for the predictions of different pre-trained models.
While focusing on this particular problem is fine, the proposed method, SRM, and the theoretical results are only loosely related to the problem: the "ensemble" part of the problem seems to be largely irrelevant.
In this respect, I feel that SRM only addresses a small issue of (robust) ensemble learning.

Another issue of the paper is the gap between the stated goal and the actual realization of SRM.
More specifically, the goal is to balance robustness and *average* performance, but the proposed method optimizes for the worst-domain performance in the vicinity of the *central* domain instead of simply the *average* domain.
No clear reason is given for why the more complicated approach is opted for.
Under the current setting, I fail to see any meaningful difference between SRM and the simpler approach using the average domain.

Empirically, SRM only marginally improves the baselines, validating my concerns above. I think the paper would greatly benefit from a better setting or scenario to demonstrate the distinct properties and effectiveness of SRM. I sincerely hope that the authors not to be discouraged if this paper is rejected because the idea of domain centrality is really interesting and I believe it will probably shine under a slightly different light.

**Claims And Evidence:**

The authors claim that the proposed method achieves superior OoD generalization compared to existing ensemble combination strategies across diverse benchmarks. However, as shown in Tables 1 and 2, the empirical improvement over ERM is marginal (66.23 $\to$ 66.54 on DomainBed, and 51.45 $\to$ 51.57 on FMoW-WILDS). Moreover, if SRM does, as the authors claim, provide a more realistic approximation of potential test distributions, then it should be better demonstrated under more general settings than ensemble learning. For example, can SRM improve the OOD performance of individual models by updating their parameters?

**Essential References Not Discussed:**

The discussion on related work is a bit lacking. A closely related work, dual risk minimization [1], also proposes to combine average risk minimization with worst-case risk minimization to mitigate the latter's over-conservativeness but does not assume multiple training distributions. Another related work is quantile risk minimization [2], which focuses on high probability domain generalization and is not mentioned in the paper either. The motivation of these papers is quite similar to the reviewed paper, although the contexts are slightly different.

[1] Li et al. "Dual Risk Minimization: Towards Next-Level Robustness in Fine-tuning Zero-Shot Models." NeurIPS 2024.
[2] Eastwood et al. "Probable domain generalization via quantile risk minimization." NeurIPS 2022.

**Experimental Designs Or Analyses:**

The experiments considered a range of baselines (both non-optimization and optimization-based ones) on common benchmarks of OOD generalization (DomainBed and WILDS). The model pool for the ensemble consists of ResNet-50 models trained with ERM under different hyperparameter settings. This is fine, but I would suggest the authors also consider some more diverse pools of models, e.g., models trained with different algorithms and/or on different datasets.

**Methods And Evaluation Criteria:**

Both the methods and the evaluation criteria make sense for the problem. The only thing that I would like to point out is that the proposed structure-informed risk minimization (SRM) is only loosely connected with ensemble learning. SRM feels like a more general principle for OOD generalization where ensemble learning is just a very narrow use case. It is unclear what consideration in SRM is specifically for ensemble learning for it to be effective.

**Other Comments Or Suggestions:**

The paper is well-written and easy to follow. I didn't find any typos or inconsistencies.

**Other Strengths And Weaknesses:**

- The centrality of a domain is an interesting and novel concept (at least to me) in the context of OOD generalization. It plays a central role in the proposed method, SRM, but I don't think it's sufficiently discussed. In particular, how does the central domain relate to the average domain? The authors seem to suggest that these two concepts are roughly equivalent, stating that SRM balances worst-case robustness with average performance, but do they really? Suppose there are two different training domains, one of which is much more likely than the other (e.g., camels are more likely to appear in deserts than on grasslands). In this case, the average domain is heavily tilted towards the more likely one, while the central domain is more like a uniform mixture of the two domains. If they are indeed different, why should one value the central domain more than the average domain?
- It is mentioned that a computationally efficient Gaussian-based approximation is used to estimate the 2-Wasserstein distance between data distributions, but many important details of this process are not provided. For example, was the distance computed over the input features or features extracted by some pre-trained model? If it was the latter case, which models were used?

**Questions For Authors:**

Please see "Other Strengths And Weaknesses".

**Relation To Broader Scientific Literature:**

I think SRM might be useful for OOD generalization in some more general settings. A lot of previous work on OOD generalization focuses on worst-case risk minimization, e.g., IRM, Group DRO, and VREx, which tend to overemphasize the importance of the worst-case domain. SRM, on the other hand, additionally considers the centrality of a domain when weighing its importance.

**Theoretical Claims:**

Proofs are not carefully checked, but the theoretical claims look sound to me.

---

> ### Author Rebuttal · Authors · 2025-04-01
>
> We thank the reviewer for the insightful feedback and acknowledging the novelty of our work. We appreciate the thoughtful comments and address them point-by-point below.
>
> **1. Why Ensemble Learning? Can SRM Improve the OOD Performance of Individual Models?**
>
> We agree that SRM is a general framework that extends beyond ensembles. Recent studies have shown that no single model can perform well across all OOD scenarios, and ensemble learning has emerged as a promising paradigm that leverages the complementary strengths of diverse models. With the growing availability of pretrained models in repositories like HuggingFace, it becomes increasingly practical and impactful to study how to effectively ensemble existing models, rather than focusing solely on training new ones.
>
> That said, SRM can indeed be applied to train individual models. To demonstrate this, we adapted SRM for single-model training and evaluated it on the PACS benchmark. Our method achieves a +0.6% average gain over CORAL, the current SOTA, showing its applicability beyond ensembles:
>
> | Method         | Art              | Cartoon           | Photo             | Sketch            | Average   |
> |----------------|------------------|--------------------|--------------------|--------------------|-----------|
> | ERM            | **88.1 (0.1)**    | 77.9 (1.3)         | 97.8 (0)           | 79.1 (0.9)         | 85.7      |
> | Group DRO      | 86.4 (0.3)        | 79.9 (0.8)         | **98.0 (0.3)**     | 72.1 (0.7)     | 84.1      |
> | CORAL (SOTA)   | 87.7 (0.6)        | 79.2 (1.1)         | 97.6 (0)           | **79.4 (0.7)**     | 86.0      |
> | SRM            | 87.6 (0.5)        | **82.0 (0.6)**     | **98.0 (0.1)**     | 78.8 (1.3)     | **86.6**  |
>
> **2. Diverse Model Pools**
>
> Following the reviewer’s suggestion, we evaluated SRM with a more diverse ensemble pool by incorporating models trained with ERM, Mixup, and CORAL. The table below shows results on TerraIncognita:
>
> | Algorithm    | L100 | L38  | L43  | L46  | Avg  |
> |--------------|------|------|------|------|------|
> | Uniform      | 50.4 | 42.6 | **62.0** | 38.7 | 48.4 |
> | ERM          | 51.3 | 43.7 | 61.4 | 39.5 | 49.0 |
> | Group DRO    | 50.9 | 43.2 | 61.8 | 39.2 | 48.8 |
> | SRM          | **52.1** | **44.3** | 61.5 | **40.0** | **49.5** |
>
> The strong result of Uniform on L43 may stem from complementary patterns captured by the diverse models. Still, SRM consistently provides better overall performance. We will include this in the revised paper. All code used is provided in the supplementary materials.
>
> **3. Related Work on Balancing Average and Worst-case Risks**
>
> We thank the reviewer for pointing out relevant related work. SRM shares the goal of balancing robustness and generalization with Dual Risk Minimization (DRM) [1] and Quantile Risk Minimization (QRM) [2], but differs in both formulation and implementation:
>
>    - DRM fine-tunes models using concept descriptions, while QRM offers probabilistic guarantees by minimizing quantile risk.
>
>    -  SRM, in contrast, introduces a structural prior derived from a distributional graph and centrality measures, offering a complementary perspective grounded in geometric relationships among training distributions.
>
> We will expand the related work section to explicitly discuss these connections in the revised manuscript.
>
> **4. Central Domain vs. Average Domain**
>
> The central domain differs from the average domain (uniform mixture) in that it emphasizes global influence within the distributional graph, not just equal weighting. For instance, in PACS (training: Art, Photo, Sketch; test: Cartoon), our graph identified Art as the most central domain. We hypothesize this occurs because "Art" inherently combines photographic elements with artistic styles, making it structurally influential.
>
> When all distributions have equal centrality, our prior reduces to a uniform prior, demonstrating that average-case optimization is a special case of SRM. This connection is discussed in lines 142–144, and empirically validated in Table 1, where SRM consistently outperforms the uniform prior across all DomainBed datasets.
>
> **5. Clarification on 2-Wasserstein Distance**
>
> The 2-Wasserstein distance used for constructing the distribution graph is computed over features extracted by the ERM-trained model with the highest validation accuracy. We will clarify this implementation detail in the revised version.
>
> We sincerely thank the reviewer again for the constructive feedback. Please let us know if you have any further questions or suggestions. We would be glad to elaborate.
>
> References:
>
> [1] Li et al. "Dual Risk Minimization: Towards Next-Level Robustness in Fine-tuning Zero-Shot Models." NeurIPS 2024.
>
>  [2] Eastwood et al. "Probable domain generalization via quantile risk minimization." NeurIPS 2022.

---

> > ### Comment · Reviewer_Qxor · 2025-04-03
> >
> > Thank you for your clear response. The added experiments look good to me. However, I still don't quite understand the motivation/utility of the central domain.
> >
> > By definition, the average performance of a model is the model's performance on the *average* domain.
> > Given that the goal is to balance robustness and average performance, my question is: why optimize the worst-case around the central domain instead of the average domain?
> > Using the PACS example, it's like: why only focus on the "Art" domain if we can train on all the domains (put another way, would the former lead to better average performance than the latter)?
> > In this context, the point that the central domain is "structurally influential" seems vacuous to me because the average domain already fully "covers" the entire graph and thus is at least as structurally influential as the central domain.
> > The paper currently lacks a clear comparison between SRM and this strong baseline using the average domain.

---

> > > ### Author Response · Authors · 2025-04-04
> > >
> > > We sincerely appreciate your thoughtful follow-up. Your question is now clearer to us, and we apologize for not addressing this point more directly in our initial rebuttal.
> > >
> > > Let us revisit your example: suppose we have two training domains—camels in deserts (majority) and camels in grasslands (minority). Since camels more frequently appear in deserts, empirical risk minimization (ERM) tends to prioritize this majority domain. Models trained with ERM generally perform well when the test domain closely resembles the training domains. We agree with the reviewer that in this scenario, the central domain coincides with the average domain. *(Note that in such cases, a meaningful distributional graph cannot be learned, and the prior defaults to a uniform distribution. Please refer to our response to Reviewer mDcw regarding this limitation of our method.)*
> > >
> > > However, the key assumption underlying our approach is that the test domain is not necessarily close to any single training domain, but is still **related** to them. This assumption is aligned with the domain adaptation theory of Ben-David et al. [1], where the H-divergence between the test and training domains is bounded by a threshold. This setting is reflected in datasets like those in DomainBed, for example:
> > > 1) PACS: Photo, Art, Cartoon, Sketch
> > > 2) TerraIncognita: images captured from distinct geographical locations.
> > >
> > > As requested by the reviewer, we have followed up on the experiments described in our rebuttal Q2 (Diverse Model Pools) and evaluated the performance when replacing the central domain prior in SRM with the average domain prior. The results are presented below:
> > >
> > > | Algorithm               | L100 | L38  | L43  | L46  | Avg  |
> > > |-------------------------|------|------|------|------|------|
> > > | Uniform | 50.4 | 42.6 | 62.0 | 38.7 | 48.4 |
> > > | ERM                     | 51.3 | 43.7 | 61.4 | 39.5 | 49.0 |
> > > | Group DRO               | 50.9 | 43.2 | 61.8 | 39.2 | 48.8 |
> > > | Average Domain Prior    | **52.1** | 43.7 | **62.3** | 39.4 | 49.4 |
> > > | SRM                     | **52.1** | **44.3** | 61.5 | **40.0** | **49.5** |
> > >
> > > We observe that in domain L43, the Average Domain Prior achieves the best performance, and in L100, it performs nearly identically to SRM (52.09% vs. 52.13%). However, SRM outperforms the average domain prior in the other two domains. We speculate that domains L43 and L100 (with accuracies of 61.5% and 52.1%) are more closely aligned with the training domains, while L46 and L38 (40% and 44.3%) are more distant.
> > >
> > > Additionally, we would like to clarify that in the PACS example, our focus is not limited to the “Art” domain. The centrality scores in this case are: Art (0.38), Photo (0.33), and Sketch (0.29). Theorem 4.2 in our paper shows that the centrality measure naturally assigns higher scores to distributions in denser regions of the distributional space.
> > >
> > > We will incorporate these insights into our revised manuscript. We sincerely appreciate the reviewer’s constructive feedback, which has helped us improve the clarity and quality of our paper. Please let us know if you have any further questions or suggestions.
> > >
> > > **Reference**
> > > [1] Ben-David, Shai, et al. *A theory of learning from different domains.* Machine Learning 79.1–2 (2010): 151–175.

---

### Decision · Program_Chairs · 2025-05-01

**Decision:**

Accept (poster)

**Comment:**

The paper proposes new approaches for Structure-informed Risk Minimization (SRM), a novel modification of the Group Distributionally Robust Optimization (Group DRO) framework, applied to robust ensemble learning. SRM improves upon Group DRO by incorporating structural information about the training domains, optimizing ensemble weights of fixed pre-trained models to minimize risk under adversarial domain mixtures centered around a data-driven “central” domain. The method is theoretically well-grounded, general in formulation, and consistently delivers performance gains across challenging datasets. Its clear design and conceptual contribution offer potential for future work in distributional robustness and out-of-distribution (OOD) generalization.

However, most of the reviewers expressed concerns about the scope and practical relevance of the method. While the paper is positioned within the context of robust ensemble learning, the actual contributions of SRM appear only tangentially connected to the ensemble aspect, focusing more broadly on risk minimization strategies. Furthermore, the gap between the paper’s stated objectives—balancing robustness and average performance—and the proposed implementation, which optimizes worst-case performance near a central domain, lacks clear justification. In addition, the added complexity over a simpler average-domain baseline is not convincingly motivated. To strengthen the work, a more compelling application scenario would be valuable in highlighting SRM’s unique strengths and necessity.